# PANTHER: Generative Pretraining Beyond Language for Sequential User Behavior Modeling

**Guilin Li**[2,*]   **Yun Zhang**[2,*]   **Xiuyuan Chen**[1,*]   **Chengqi Li**[1,3]   **Bo Wang**[2]

**Linghe Kong**[1]   **Wenjia Wang**[5]   **Weiran Huang**[1,3,†]   **Matthias Hwai Yong Tan**[4]

[1]Shanghai Jiao Tong University   [2]WeChat Pay, Tencent

[3]Shanghai Innovation Institute [4]City University of Hong Kong

[5]Hong Kong University of Science and Technology (Guangzhou)

## Abstract

Large language models (LLMs) have shown that generative pretraining can distill vast world knowledge into compact token representations. While LLMs encapsulate extensive world knowledge, they remain limited in modeling the behavioral knowledge contained within user interaction histories. User behavior forms a distinct modality, where each action—defined by multi-dimensional attributes such as time, context, and transaction type—constitutes a behavioral token. Modeling these high-cardinality, sparse, and irregular sequences is challenging, and discriminative models often falter under limited supervision. To bridge this gap, we extend generative pretraining to user behavior, learning transferable representations from unlabeled behavioral data analogous to how LLMs learn from text. We present PANTHER, a hybrid generative–discriminative framework that unifies user behavior pretraining and downstream adaptation, enabling large-scale sequential user representation learning and real-time inference. PANTHER introduces: (1) Structured Tokenization to compress multi-dimensional transaction attributes into an interpretable vocabulary; (2) Sequence Pattern Recognition Module (SPRM) for modeling periodic transaction motifs; (3) a Unified User-Profile Embedding that fuses static demographics with dynamic transaction histories, enabling both personalized predictions and population-level knowledge transfer; and (4) Real-time scalability enabled by offline caching of pre-trained embeddings for millisecond-level inference.Fully deployed and operational online at WeChat Pay, PANTHER delivers a 25.6% boost in next-transaction prediction HitRate@1 and a 38.6% relative improvement in fraud detection recall over baselines. Cross-domain evaluations on public benchmarks (CCT, MBD, MovieLens-1M, Yelp) show strong generalization, achieving up to 21% HitRate@1 gains over transformer baselines, establishing PANTHER as a scalable, high-performance framework for industrial user sequential behavior modeling.

## 1  Introduction

Online payment platforms, such as WeChat Pay, Alipay, and PayPal, process billions of transactions daily, supported by critical applications like fraud detection, credit risk assessment, and personalized marketing [1]. Modeling payment behavior at this scale is challenging: data volumes are extreme; categorical features (e.g., merchant category, payment channel) are high-cardinality; and real-time risk decisions must be delivered under 100 ms.

---

[*]Equal contribution. Contact: {guilinli,aaayunzhang}@tencent.com.

[†]Corresponding author. Contact: weiran.huang@outlook.com.

39th Conference on Neural Information Processing Systems (NeurIPS 2025).

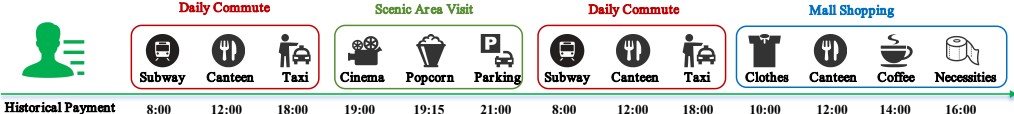

Figure 1: Illustration of the periodic pattern of user behaviors

Recent advances in self-supervised pretraining have transformed representation learning in language, vision, and recommendation [2–6]. Large language models (LLMs) demonstrate that generative pretraining on unlabeled text can compress extensive *world knowledge* into token representations. Payment platforms, however, require models that capture *behavioral knowledge*—the individualized regularities, intents, and deviations embedded in users' interaction histories. In this modality, each action is a structured event rather than a word; we therefore view a transaction as a *behavioral token* defined by multi-dimensional attributes (time, context, device, counterparty, amount). Extracting user-relevant information from these high-cardinality, sparse, and irregular sequences is essential for understanding users and for operational decision-making at scale.

Mainstream industrial approaches, adapted from recommendation systems (e.g., DeepFM [7], DCN [8], AutoFIS [9], DIEN [10]), rely primarily on supervised discriminative models for transaction-level classification. At billion-user scale they face four persistent limitations: (1) *label scarcity*—positive examples are too few to span the combinatorial feature space; (2) *overfitting to high-dimensional categories*—models memorize spurious co-occurrences rather than meaningful risk patterns; (3) *latency-driven truncation* of long histories—weakening the ability to leverage long-range behavior; and (4) *static embeddings*—biased toward frequent labels and brittle under cold-start and long-tail distributions [11].

Although sequential recommenders increasingly adopt generative objectives, they typically use generation as an *end task* (next-item prediction) (for example HSTU [6]) rather than as a *pretraining mechanism* to compress *user knowledge* into transferable representations. In contrast, we adopt a *pretrain→adapt* perspective: learn generalizable user embeddings from unlabeled behavioral logs *offline*, then *online* adapt them with lightweight discriminative heads to satisfy production latency constraints. This hybrid generative–discriminative design targets *user understanding* rather than only item generation, and supports multiple downstream decisions (fraud detection, transaction prediction, recommendation).

Orthogonal to this objective/framework difference, user behavior sequence data inherently differs from other sequential modalities. Unlike natural language, which is governed by grammatical structures, user behavior sequences exhibit rich recurring patterns—daily routines, weekly cycles, and seasonal trends—reflecting habitual user behaviors (illustrated in Figure 1). Standard sequential architectures, including Transformers, process events individually through self-attention mechanisms. Although powerful, these methods inadequately capture local periodicities and global relational patterns inherent to payment data, often diluting signals from lengthy sequences and missing subtle yet crucial anomalies indicative of fraud.

We propose **PANTHER** (Pattern AttentioN Transformer with Hybrid User ProfilER), a hybrid generative–discriminative framework that unifies *user behavior pretraining* with *downstream adaptation* for real-time inference. *Offline*, a PANTHER transformer is pretrained on billions of transactions to predict subsequent events, producing compact user-profile embeddings and event likelihoods that encode long-term intent and temporal dynamics. *Online*, a lightweight classifier fuses these cached representations with current transaction context to compute risk within milliseconds. This design leverages generative pretraining for representational power while keeping inference efficient for high-throughput systems.

To make pretraining effective on user behavior sequences and inference feasible online, PANTHER introduces three modeling components and one systems mechanism:

1. **Token Space Compression.** A *structured tokenization* scheme integrates contextual and counterparty attributes into unified tokens and applies frequency-aware compression to reduce dimensionality, filtering noise and stabilizing generative learning over heterogeneous inputs.

2. **Pattern-Aware Convolutional Cross-Attention.** A *Sequence Pattern Recognition Module (SPRM)* blends multi-scale (depthwise) convolutions with cross-attention to capture local periodicities (e.g., daily/weekly cycles) together with broader contextual dependencies—preserving cyclical signals while maintaining global relations.

3. **User Profile Embedding with Contrastive Personalization.** A dedicated *user-profile token* provides persistent access to user context across the sequence; a contrastive objective arranges users with similar payment behaviors nearby in latent space, improving personalization under sparsity and cold-start.

4. **Real-Time Hybrid Inference.** Pretrained user/profile embeddings are cached offline and fused online with context, recent patterns, and deviation features to produce millisecond-level posterior scores—meeting production latency constraints.

We empirically validate PANTHER on real-world WeChat Pay data, demonstrating strong generalization across fraud detection, transaction prediction, personalized user modeling, and recommendation. PANTHER yields a 25.6% improvement over Transformer baselines on internal WeChat Pay benchmarks and a 21% HR@1 gain on MovieLens-1M; on Yelp, it improves NDCG@5 by 29.6% over DCN. A production PANTHER-based fraud system at WeChat Pay improves Top-0.1% recall by 38.6% in online A/B tests, enhancing security across billions of daily transactions.

In summary, PANTHER provides a scalable, efficient approach to modeling complex sequential user behaviors by extending generative pretraining beyond language to the behavioral modality, compressing user knowledge into transferable representations, and bridging pretraining with real-time inference for industrial sequential decision-making.

## 2 Related Work

**Sequential Deep Learning and Generative Recommendation.** Deep learning methods for sequential modeling have significantly advanced recommendation and personalization, evolving from early RNN- and CNN-based models (e.g., GRU4Rec [12], Caser [13]) to recent transformer-based approaches (e.g., SASRec [14], BERT4Rec [15]). Generative sequential models, such as HSTU [6], TIGER [16], DiffuRecSys [17], and HLLM [18] have further advanced the field by modeling complex temporal dependencies and uncertainties. Despite these advancements, several aspects remain under-explored: explicit modeling of periodic behaviors (e.g., daily or weekly patterns), dedicated long-term personalized user embeddings, and computational strategies for low-latency inference. PANTHER uniquely addresses these challenges by incorporating convolutional cross-attention for periodic behavior modeling, contrastive user personalization embeddings, and cached representations enabling efficient real-time inference.

**Fraud Detection in Financial Systems.** Fraud detection in financial systems involves severe class imbalance, label scarcity, and real-time inference constraints. Early supervised approaches, including logistic regression, decision trees, and gradient boosting, perform effectively on structured data but face challenges with sparse, noisy, high-dimensional transaction data and extreme class imbalance [19]. Graph-based methods like R-GCNs [20] and heterogeneous GNNs [21] effectively capture relational patterns among entities, yet scalability and real-time latency in billion-user scenarios remain open research areas. PANTHER complements these methods by leveraging transformer-based generative pretraining, efficiently modeling temporal user behaviors and maintaining millisecond-level inference.

## 3 PANTHER

### 3.1 Model Overview and Problem Formulation

PANTHER employs a two-stage architecture for payment fraud detection, combining offline generative pretraining with real-time inference. Let $\mathcal{U}$ denote our user base where each user $u \in \mathcal{U}$ generates a sequence of payment events $\mathbf{X}_u = [x_1, x_2, \ldots, x_L]$ with $x_t \in \mathcal{V}$ representing compressed transaction tokens (see §3.2). Our system aims to estimate the fraud probability:

$$Pr(y = 1 \mid x_{\text{new}}, \mathbf{c}_{\text{new}}, \mathbf{X}_u), \tag{1}$$

for each new transaction $x_{\text{new}}$ with contextual features $\mathbf{c}_{\text{new}}$, given the user's historical sequence $\mathbf{X}_u$.

**Stage 1: Pretraining for Next Transaction Prediction.** We first learn user behavior patterns through next payment behavior prediction. Each user behavior sequence is augmented with a learnable profile token $x_{\text{profile}}$ encoding static attributes (see §3.4). Our transformer-based model with SPRM

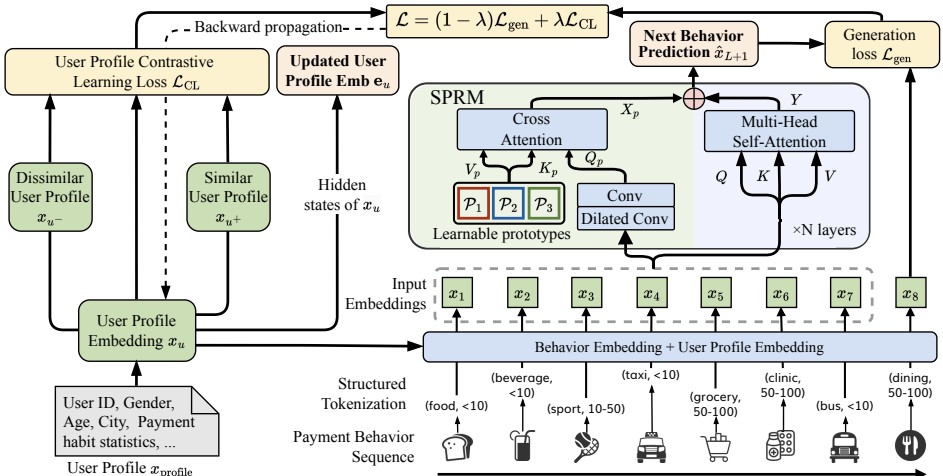

Figure 2: Key components of PANTHER: Structured Tokenization, SPRM and user profile embedding modules ( §3.3) optimizes:

$$\mathcal{L}_{\text{gen}} = \mathbb{E}_{u \sim \mathcal{U}} \left[ \sum_{t=1}^{L} -\log Pr_\theta(x_t \mid x_{<t}, x_{\text{profile}}) \right], \tag{2}$$

where the loss is the negative log-likelihood of the ground-truth next behaviors over the sequence. This produces two key pretrained outputs: (1) user profile embeddings $\mathbf{e}_u \in \mathbb{R}^d$, and (2) behavior predictors $\mathbf{X}_u$ that predict next behavior $Pr_\theta(\hat{x}_{L+1}|\mathbf{X}_u)$ through a linear and softmax layer.

**Stage 2: Hybrid Inference for Real-Time Fraud Detection.** For live transactions, we compute risk scores through feature fusion:

$$Pr(y = 1|\cdot) = g_\phi \left( \underbrace{\psi(\mathbf{c}_{\text{new}})}_{\text{context}}, \underbrace{f_{\text{enc}}(\mathbf{X}_u^{[L\text{-}100:L]})}_{\text{recent patterns}}, \underbrace{\mathbf{e}_u}_{\text{long-term profile}}, \underbrace{\Delta(\hat{x}_{L+1}, x_{\text{new}})}_{\text{behavior deviation}} \right), \tag{3}$$

where the function $g_\phi$ represents a general downstream discriminative model, which can take various flexible forms depending on specific tasks or use cases, $\psi(\cdot)$ embeds transaction context features, $f_{\text{enc}}$ encodes the last 100 transactions via TextCNN [22], and $\Delta$ measures the distance between predicted and observed behaviors.

This design strikes a balance between accurate long-term behavior modeling and responsive real-time decision-making, making it particularly well-suited for high-throughput fraud detection systems.

## 3.2 Structured Tokenization for Payment Behaviors

Unlike natural language, payment behaviors lack predefined semantic units, as each transaction is defined by a combination of *contextual features* (e.g., payment channel, discretized amount) and *counterparty attributes* (e.g., merchant category, risk level). To efficiently capture this structured, multi-dimensional information, we introduce a *structured tokenization* framework that integrates these attributes into a unified representation.

Each transaction token is formed as the Cartesian product of contextual and counterparty features:

$$\tau = (c_i, a_j, m_k, r_l) \in \mathcal{C} \times \mathcal{A} \times \mathcal{M} \times \mathcal{R}, \tag{4}$$

where, for instance, $c_i$ might represent a payment channel such as `CreditCard` or `RedPocket`, and $a_j$ could capture the transaction amount, discretized into ranges like `$10-50` and `$50-100`. On the counterparty side, $m_k$ can capture the merchant category, while $r_l$ indicate the associated risk level, reflecting the merchant's historical reliability (e.g., `HighRisk`, `LowRisk`). This tokenization approach captures the key transactional semantics by transforming raw payment behaviors into compact, domain-specific tokens (e.g., (e.g., `CreditCard_$50-100_Fuel_LowRisk`).), effectively embedding transactions within a structured, context-rich feature space. This design allows the

pretraining model to learn meaningful representations of transaction behavior, preserving critical financial signals while reducing sparsity.

However, the raw combinatorial space is $|\mathcal{V}| = |\mathcal{C}| \times |\mathcal{A}| \times |\mathcal{M}| \times |\mathcal{R}| \approx 2\text{M}$, which is prohibitively large, leading to severe sparsity and overfitting risks. To address this, we adopt a *frequency-based compression*, leveraging real-world transaction distributions to retain only the top $K = 60,000$ most frequent tokens, covering over 96% of historical transactions. Less frequent, long-tail combinations are mapped to a unified [UNK] token, effectively reducing the vocabulary size by 97% (from 2M to 60k), while preserving high-impact and interpretable patterns. It maintains critical transaction semantics while significantly reducing the model's computational footprint.

## 3.3 Sequence Pattern Recognition Module (SPRM)

Payment sequences, unlike natural language, lack a formal grammar, yet exhibit structured recurring patterns, such as sequential routines and periodic behaviors (as illustrated in Figure 1). Accurately modeling these patterns is critical for applications like next-payment prediction and fraud detection, as they encapsulate context-aware user habits. However, standard self-attention in Transformers processes each event independently, incurring a quadratic complexity and lacking inductive biases for local and periodic structures. This inefficiency can dilute signals in long sequences and obscure subtle deviations from the routine. To address these limitations, we introduce the Sequence Pattern Recognition Module (SPRM), which incorporates two lightweight inductive biases to explicitly capture local and periodic patterns in payment sequences. Operating in parallel with the multi-head self-attention, the SPRM enhances the final representation by adding its output to the Transformer's output, forming a composite result that leverages both global context and structured pattern recognition.

**(i) Local Pattern Aggregation.** To capture the multi-scale nature of transactional routines, we apply depthwise dilated convolutions to the token embeddings $H \in \mathbb{R}^{T \times d}$, using a range of kernel sizes $w$ and dilation rates $r$:

$$H_p^{(k)} = \text{Conv}_{\text{dil}=r_k,\, w=w_k}(H), \quad H_p = \text{Concat}_k\big(H_p^{(k)}\big).$$

Here, kernels with smaller dilation rates (e.g. $w = 3$, $r = 1$) capture immediate temporal clusters, while larger dilated kernels (e.g., $w = 3$, $r = 2$) capture periodic or recurring patterns, even in the presence of occasional noise or sporadic behaviors. This convolutional aggregation efficiently captures multiscale transactional patterns in linear time, providing a compact representation that encodes both short-term and long-term transactional routines while remaining robust to random fluctuations within those patterns.

**(ii) Prototype-Driven Pattern Matching.** To further enrich these multiscale embeddings, we introduce $m$ learnable prototypes $\mathcal{P} \in \mathbb{R}^{m \times d}$, each representing a canonical spending motif (e.g., *weekend leisure*, *weekday commute*). These prototypes act as structured anchors in the embedding space, providing a scaffold for the model to map observed behaviors to known patterns.

$$Q = H_p W_Q, \quad K = \mathcal{P} W_K, \quad V = \mathcal{P} W_V, \quad X_p = \text{Softmax}\left(\frac{QK^\top}{\sqrt{d_k}}\right)V.$$

This cross-attention mechanism compresses each local segment onto the nearest prototypes, effectively aligning real-world sequences to interpretable, high-level motifs, while preserving local context. Unlike vanilla self-attention, which treats every token as context-free, this approach enforces a structured alignment, encouraging the model to form sparse, interpretable codes that highlight subtle departures from routine behavior.

**Complexity Analysis.** For a sequence of length $T$, standard self-attention incurs a quadratic $\mathcal{O}(T^2)$ complexity. In contrast, SPRM introduces only a linear cost for dilated convolutions, $\mathcal{O}(T)$, and a modest $\mathcal{O}(Tm)$ for prototype cross-attention, where $m \ll T$ (typically $m = 64$ in practice). This results in an overall complexity of approximately $\mathcal{O}(T)$, making it feasible for long payment sequences without sacrificing long-range dependency capture.

## 3.4 User Profile Embedding for Personalized Payment Predictions

In PANTHER, we propose a novel *user profile embedding* that enables personalized transaction predictions by learning shared behavioral patterns among demographically similar users. This embedding combines static user attributes with dynamic transaction histories to form a compact latent

Table 1: Next-transaction prediction results on WeChat Pay. Relative improvements reflect PANTHER's gains over the Transformer baseline.

| Method | HR@1 | HR@5 | HR@10 |
|---|---|---|---|
| Transformer | 0.1952 | 0.4121 | 0.5308 |
| SASRec | 0.2041 | 0.4280 | 0.5347 |
| HSTU | 0.2089 | 0.4271 | 0.5320 |
| PANTHER (SPRM only) | 0.2243(+14.9%) | 0.4493(+9.03%) | 0.5421(+2.13%) |
| + Profile as First Token | 0.2301(+17.9%) | 0.4634(+12.45%) | 0.5468(+3.01%) |
| + Profile as Positional Encoding | 0.2351(+20.4%) | 0.4774(+15.85%) | 0.5568(+4.90%) |
| + Profile + CL | **0.2452 ± 0.0005 (+25.6%)** | **0.4837 ± 0.0009 (+17.37%)** | **0.5647 ± 0.0012 (+6.39%)** |

representation, which simultaneously serves two purposes: (1) as a personalized positional encoding that contextualizes user-specific transaction sequences, and (2) as a learnable similarity anchor that adaptively refines historical behavior patterns through contrastive learning.

Our contrastive objective follows an information-theoretic formulation:

$$\mathcal{L}_{\text{CL}} = - \sum_{(i,j) \in Pos} \log \frac{\exp(-\|e_i - e_j\|_2 / \tau)}{\sum_{k \in \mathcal{U} \setminus \{i\}} \exp(-\|e_i - e_k\|_2 / \tau)}, \quad (5)$$

where $Pos$ denotes positive pairs of users sharing demographic attributes (age$\pm 2$, same geographic region, etc.), $\mathcal{U}$ represents the user population, and $\tau$ controls the similarity concentration temperature. This objective maximizes mutual information between demographically similar users while maintaining separation from dissimilar counterparts through the denominator's hard negative mining over all non-positive pairs.

The complete optimization objective combines both components:

$$\mathcal{L} = \underbrace{(1 - \lambda)\mathcal{L}_{\text{gen}}}_{\text{Individual fidelity}} + \underbrace{\lambda \mathcal{L}_{\text{CL}}}_{\text{Population structure}}. \quad (6)$$

This dual-objective formulation yields three key advantages: (1) geometrically meaningful embeddings where user similarity correlates with both demographic alignment and behavioral consistency, (2) improved sample efficiency through knowledge transfer between similar users, and (3) inherent regularization that prevents overfitting to individual transaction outliers.

# 4 Experiments

## 4.1 Real-World Deployment & Performance Validation

We validate the core contributions of PANTHER through its large-scale deployment at WeChat Pay, focusing on two key aspects: pretraining efficacy (next-transaction prediction) and downstream fraud detection capabilities. This deployment addresses the challenges identified in Section 3.1, showcasing how PANTHER can enhance fraud detection and personalized user services in real-world setting.

### 4.1.1 Next-Transaction Prediction Benchmark

**Task & Dataset:** models are pretrained on 5.3B anonymized transactions (38M users over 6 months) to model user-specific behavior. The raw transactions data, consisting six attributes (amount, merchant category, etc.), are tokenized by 60k interpretable tokens with the structured tokenization scheme.

**Experimental Setup:** In the next-transaction prediction task, we evaluate PANTHER's ability to predict the next transaction based on a user's historical data. The model leverages the SPRM and unified user-profile embeddings for this purpose. We employ two widely-adopted evaluation metrics: HR@K (Hit Ratio at K) and NDCG@K (Normalized Discounted Cumulative Gain at K). Specifically, HR@K measures the fraction of test instances in which the ground-truth item appears among the top-K predicted items. NDCG@K assesses the ranking quality by assigning higher weights to relevant items placed at top positions, normalized by the ideal discounted gain. The code is available at https://github.com/WeChatPay-Pretraining/PANTHER.

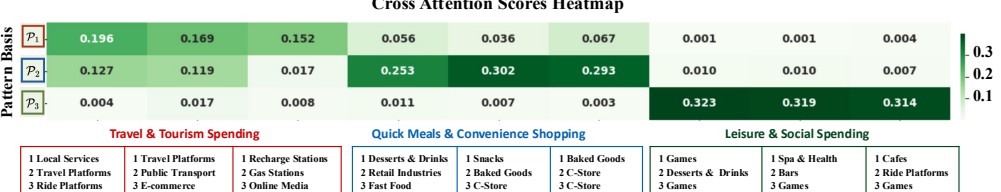

Figure 3: Demonstration of the cross attention score between recurring three-gram user payment behaviors and the learnable pattern prototypes.

We benchmark PANTHER against several strong baseline models, including Transformer [23], SASRec [14], and HSTU [6].

**Key Findings**: As shown in Table 1, PANTHER outperforms baseline models significantly. The SPRM module alone improves HR@1 by 14.9%, highlighting the value of sequential behavior patterns. Incorporating user profiles via learnable positional encoding boosts HR@1 by an additional 5.5%. Adding contrastive learning objectives results in a total HR@1 improvement of 25.6%, demonstrating the effectiveness of context-aware embeddings and knowledge transfer.

**Visualization of Learned Patterns:** Figure 3 shows how PANTHER maps consecutive user behaviors to learned behavior prototypes (P1, P2, P3). Each column represents a sequence of three consecutive behaviors identified as recurring patterns in the user's history. The heatmap values indicate the strength of alignment between behaviors and prototypes, with higher values signifying stronger matches. This visualization highlights how the model captures and recognizes repeating transaction motifs over time.

### 4.1.2 Hybrid Real-Time Fraud Detection

We evaluate PANTHER's fraud detection performance through a 10-day full-traffic A/B test on WeChat Pay's production system, using the DeepFM model [7] as the baseline. DeepFM relies on handcrafted features and a TextCNN encoder for sequence processing. PANTHER operates in a hybrid configuration, combining offline-pretrained user-profile embeddings ($e_u$) and behavioral anomaly scores ($\Delta$) with real-time transaction features (Equation 3), achieving substantial recall improvements, as shown in Figure 4.

The hybrid PANTHER model improves fraud recall by 109.5% at the 0.01% threshold, with smaller gains at higher thresholds (0.1% **+38.6%**, 1% +12.1%). The larger gains at extreme thresholds highlight the model's effectiveness in detecting low-frequency, high-risk fraud cases using personalized embeddings ($e_u$) and behavioral anomaly scores ($\Delta$).

**Deployment Strategy:** This hybrid approach introduces minimal overhead (5ms higher than the baseline), while offering three key advantages: (1) personalized fraud detection via user-specific embeddings, (2) explainable anomaly detection through interpretable deviation scores, and (3) scalable production deployment by separating compute-intensive pretraining from real-time inference.

### 4.1.3 Merchant Risk Assessment via Behavior Sequence Pretraining

In addition to fraud detection, we developed a framework for merchant risk assessment based on behavior sequence pretraining. This approach identifies merchants potentially involved in fraud by analyzing deviations in user behavior at the merchant level, incorporating components for behavior prediction, deviation measurement, risk aggregation, and merchant classification.

**Next Behavior Deviation Scoring.**     Deviation is measured as the standardized difference between the predicted likelihood of a behavior and the user's typical behavior distribution. It reduces false positives for users with naturally varied transaction patterns: $\Delta_{u,m} = \frac{P_u(m)-\mu_u}{\sigma_u}$, where $\mu_u$ and $\sigma_u$ are the mean and Std Dev of predicted probabilities across all potential behaviors for user $u$.

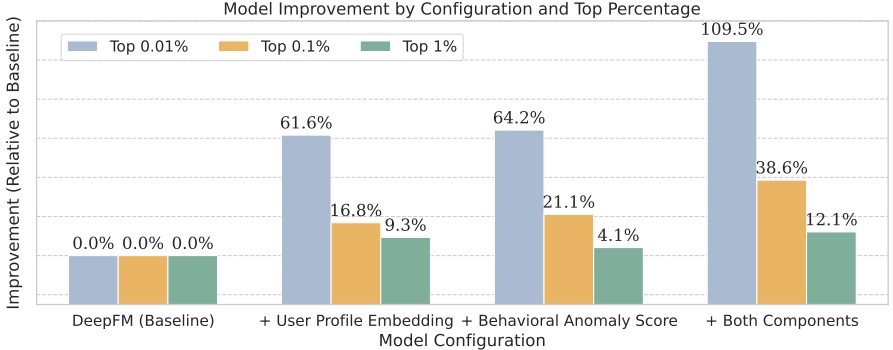

Figure 4: Fraud recall improvement at operational thresholds (Recall@Top-K).

**Merchant Risk Scoring and Classification.** We aggregate deviation scores for all users transacting with merchant $m$ and compute the quantile-based risk score:

$$R_m = \text{Quantile}_q(\{\Delta_{u,m} \mid u \in \mathcal{U}_m\}),$$

where $\mathcal{U}_m$ is the set of users interacting with merchant $m$. Robust statistical features are extracted and used as inputs for classifier (e.g., XGBoost) to label merchants as high or low risk.

**Practical Impact.** The framework showed strong performance in practice, achieving 76% accuracy for flagged high-risk merchants – a significant improvement over the baseline method's 50% accuracy.

## 4.2 Benchmark Performance

After validating PANTHER's real-world viability, we evaluate its performance on public benchmark datasets to quantify its improvements over existing methods. This section compares PANTHER against strong baselines on public transaction datasets and recommendation benchmark, demonstrating consistent performance gains across domains beyond the WeChat Pay setting.

### 4.2.1 Datasets & Tasks

We evaluate PANTHER's pretraining performance on four datasets, including two financial datasets and two recommendation datasets, to assess its effectiveness and generalizability. For downstream tasks, we compare fraud detection performance on CCT and recommendation tasks on Yelp (since only these datasets include user profile features).

1. **Credit Card Transactions (CCT)** [24]: A synthetic dataset with 20 million transactions from 2,000 users, used for fraud detection with embedded anomalies.
2. **MBD-mini** [25]: An anonymized banking dataset tracking monthly product purchases, focusing on repetitive consumer behaviors.
3. **MovieLens-1M** [26]: A widely-used recommendation dataset with 1 million user-item interactions, ideal for evaluating sequential models in non-financial domains.
4. **Yelp** [27]: A public recommendation dataset containing millions of user-business interactions (e.g., restaurant reviews), commonly used for sequential recommendation tasks.

Each dataset is split chronologically into training, validation, and test subsets.

**Implementation details:** We apply the same model hyperparameters to the open benchmark datasets as those used for the WeChat Pay dataset. Full implementation details are provided in Section B and our supplementary code.

### 4.2.2 Next Payment Behavior Prediction Benchmarking

We assess PANTHER on next behavior prediction. As summarized in Table 2 for MBD-mini, CCT, MovieLens-1M and Yelp, it consistently outperform strong baselines like SASRec and HSTU.

Table 2: Experimental results of various methods on transaction and recommendation datasets.

| | Dataset | Method | HR@1 | HR@5 | HR@10 |
|---|---|---|---|---|---|
| Payment | MBD-mini | Transformer | .0441 | .1735 | .2864 |
| | | SASRec | .0442(+0.23%) | .1729(-0.35%) | .2871(+0.24%) |
| | | HSTU | .0432(-2.04%) | .1713(-1.27%) | .2851(-0.45%) |
| | | PANTHER | **.0454(+2.95%)** | **.1766(+1.79%)** | **.2923± .0005(+2.06%)** |
| | CCT | Transformer | .0500 | .1381 | .1979 |
| | | SASRec | .0447(-10.60%) | .1387(+0.43%) | .1994(+0.76%) |
| | | HSTU | .0537(+7.40%) | .1419(+2.75%) | .2025(+2.32%) |
| | | PANTHER | **.0576(+15.20%)** | **.1554(+12.53%)** | **.2176± .0004(+9.95%)** |
| Recommendation | MovieLens-1M | Transformer | .0579 | .1826 | .2773 |
| | | SASRec | .0627(+8.29%) | .1906(+4.38%) | .2853(+2.88%)[*] |
| | | HSTU | .0699(+20.73%) | .1972(+7.99%) | .3043(+9.74%) |
| | | PANTHER | **.0705(+21.76%)** | **.2103(+15.17%)** | **.3078± .0007(+11.01%)** |
| | Yelp | Transformer | .0851 | .1769 | .2300 |
| | | SASRec | .0875(+2.74%) | .1820(+2.91%) | .2385(+3.71%) |
| | | HSTU | .0879(+3.21%) | .1878(+6.19%) | .2496(+8.55%) |
| | | PANTHER | **.0929(+9.17%)** | **.2204(+24.63%)** | **.2924± .0006(+27.14%)** |

[*] The higher HR@10 reported SASRec [14] is due to sampled negative evaluation (∼100 items). Our protocol follows HSTU with full-ranking over ∼3,700 items.

Table 3: Recommendation Performance on Yelp

| Model | HR@1 | HR@5 | NDCG@5 |
|---|---|---|---|
| DCN | 0.612 | 0.963 | 0.534(baseline) |
| + PANTHER | 0.773 | 0.982 | 0.692(+29.6%) |

Table 4: Fraud detection performance on CCT

| Model | Recall | Accuracy | F1 Score |
|---|---|---|---|
| TabBERT-MLP | - | - | 0.760 (baseline) |
| TabBERT-LSTM | - | - | 0.860 (+13.2%) |
| DCN | 0.931 | 0.871 | 0.888 (+ 16.8% ) |
| + PANTHER | 0.978 | 0.896 | 0.911 (+ 19.9%) |

**Key Observations:** On WeChat Pay, PANTHER-large achieves a 25.56% HR@1 improvement over Transformer, reflecting its capability to model sporadic, large-scale financial transactions. On MBD-mini and CCT, PANTHER improves HR@1 by 2.95% and 15.2%, respectively, demonstrating its broad applicability to other payment transaction data. On the MovieLens-1M and Yelp datasets, PANTHER improves HR@1 by 21.8% and 9.17% over Transformer, surpassing HSTU and showing strong generalization beyond payment data. Overall, PANTHER demonstrates robustness across domains and highlights the advantages of larger model configurations for complex user-item interactions.

### 4.2.3 Hybrid Fraud Detection & Recommendation

We evaluate the PANTHER on downstream fraud detection and recommendation tasks, by introducing its pretrained embeddings ($e_u$) and next-behavior predictions ($\hat{X}_t$) to the baseline models.

**Hybrid Recommendation on Yelp.** We demonstrate that PANTHER not only excels in fraud detection but also performs effectively in recommendation tasks. For example, on the Yelp dataset, the NDCG@5 metric improves by 29.6% over the DCN baseline (Table 3), highlighting the value of pretrained embeddings in enhancing recommendation performance.

**Hybrid Fraud Detection on CCT.** On the CCT dataset, incorporating PANTHER's pretrained embeddings and predictions improves fraud detection recall by 19.9% over the TabBERT-MLP ([28]) baseline (Table 4), enhancing the model's effectiveness in detecting fraudulent activity.

These results confirm that PANTHER's hybrid method significantly boosts performance across tasks by leveraging learned user profiles and behavior predictions.

### 4.3 Transferability

We examine PANTHER's ability to transfer learned representations across new users, datasets, and domains, demonstrating the power of generative pretraining in low-label and cross-domain scenarios. PANTHER shows exceptional transferability, with a 301.4% improvement over cold-start baselines when transferring across user demographics. Additionally, fine-tuning on external datasets after pretraining on WeChat Pay leads to an average HR@1 improvement of 16.66% on MBD-mini and CCT, showcasing the model's adaptability across diverse transaction contexts. More detailed results and experiments are provided Section C. These findings highlight PANTHER's ability to generalize effectively with minimal retraining, making it highly suitable for real-world applications with sparse labeled data.

### 4.4 Summary of Experimental Findings

Our experiments show that PANTHER consistently outperforms strong baselines across tasks. It achieves robust next-transaction prediction with noisy, sparse data by leveraging the Sequence Pattern Recognition Module and adaptive user embeddings. PANTHER also demonstrates strong transferability, excelling across diverse domains, including transaction data (CCT, MBD-mini) and recommendation tasks (MovieLens-1M, Yelp). Real-world deployment at WeChat Pay shows a 38% improvement in fraud recall at top 0.1%. Ablation studies confirm that key components, like SPRM and contrastive learning, significantly enhance performance. Overall, it proves to be a versatile, scalable solution for sequential behavior modeling, with strong generalization across domains.

## 5 Conclusions

We introduced PANTHER, a generative pretraining framework that addresses real-world payment data complexities by combining noise suppression, pattern recognition, and user personalization. Through token space compression and innovative attention mechanisms, PANTHER uncovers subtle, cyclic behaviors and incorporates long-term user context, producing high-fidelity user embeddings. This design supports critical financial applications such as fraud detection, credit scoring, next-payment prediction, and user segmentation. Beyond industry use, PANTHER highlights an important direction for the machine learning community: leveraging massive unlabeled transaction logs to enhance efficiency and adaptability. As financial and e-commerce data volumes rise, PANTHER's integration of generative modeling and personalization enables more secure, accurate, and user-centric services. However, its lack of interpretability is a limitation, as complex representations may hinder transparency in high-stakes applications. Future work will focus on improving explainability for broader applicability in regulated domains.

## Acknowledgments

This work is supported by the National Natural Science Foundation of China (No. 62406192), Opening Project of the State Key Laboratory of General Artificial Intelligence (No. SKLAGI2024OP12), and Tencent WeChat Rhino-Bird Focused Research Program.

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

# Appendix

## A  Notation Table

We summarize the frequently used notations in Table 5.

Table 5: Notations used in this paper

| Notation | Descrpition |
|---|---|
| $\mathbf{X}_u = [x_1, x_2, \ldots, x_L]$ | The payment behavior sequence of length $L$ for a user $u \in \mathcal{U}$. |
| $x_t \in \mathcal{V}$ | The transaction token at time step $t$, $\mathcal{V}$ is the set of possible transaction tokens. |
| $P(y = 1 \mid x_{\text{new}}, \mathbf{c}_{\text{new}}, \mathbf{X}_u)$ | Fraud probability for new transaction $x_{\text{new}}$ with contextual feature $\mathbf{c}_{\text{new}}$, given $\mathbf{X}_u$. The $y$ is an indicator variable for fraud. |
| $\mathcal{L}_{\text{gen}}$ | The generative loss function for next behavior prediction |
| $P_\theta(x_t \mid x_{<t}, x_{\text{profile}})$ | The probability of the $t$-th transaction $x_t$ given the previous transactions $x_{<t}$, user profile features $x_{\text{profile}}$ and model parameters $\theta$. |
| $e_u$ | The learned user profile embedding for user $u$. |
| $g_\phi(\cdot)$ | Real-time fraud detection model that integrates features including recent patterns, predicted deviations, and user profiles for risk prediction, typically a DCN network. |
| $\psi(\cdot)$ | The embedding function for transaction context features, typically in the form of a linear layer. |
| $f_{\text{enc}}(\cdot)$ | Sequence encoder for user's recent short-term payment behaviors, typically a TextCNN model. |
| $\Delta(x_{\text{new}}, \hat{x}_{L+1})$ | The deviation between the observed new transaction $x_{\text{new}}$ and the predicted next behavior $\hat{x}_{L+1}$. |
| $\tau = (c_i, a_j, m_k, r_l) \in \mathcal{C} \times \mathcal{A} \times \mathcal{M} \times \mathcal{R}$ | Example definition of transaction token formed by the Cartesian product of the contextual and counterparty features. |
| $\mathrm{P}_{\text{SPRM}} \in \mathbb{R}^{m \times d}$ | The set of $m$ learnable prototypes of embedding dimension $d$ in the Sequence Pattern Recognition Module (SPRM). |
| $H \in \mathbb{R}^{T \times d}$ | Token embeddings of $d$ dimension for an input sequence of length $T$ |
| $\mathcal{L}_{\text{CL}}$ | The contrastive loss function for user profile embedding learning |
| $(i, j) \in P_{pos}$ | positive pairs of users sharing similar demographic attributes |
| $e_i, e_j$ | User profile embeddings for users $i$ and $j$, respectively. |
| $\lambda$ | The hyper-parameter balancing the two losses: $\mathcal{L}_{\text{gen}}, \mathcal{L}_{\text{CL}}$ |
| $R_m$ | The risk score for merchant $m$, computed from the user deviation scores interacting with the merchant. |

# B Implementation Details

In this section, we provide the full implementation details for MBD-mini, CCT, Movielens, and Yelp datasets, including model configurations, training procedures, and tokenization methods.

## B.1 CCT, Yelp, and MBD-mini Experiments

For the CCT, Yelp, and MBD-mini dataset, we configured PANTHER with **4 layers** and **2 attention heads**. Training was conducted with a **batch size of 128** at learning rate $1 \times 10^{-3}$. The training utilized a single GPU over a span of 2 hours on CCT, 6 hours on Yelp, and 12 hours hours on MBD-mini.

The Transformer, SASRec, and HSTU models were configured with the same learning rate, number of layers, and batch size as PANTHER.

## B.2 MovieLens-1M Experiments

For the MovieLens-1M experiments, PANTHER was trained with a **batch size of 128** and a learning rate of $1 \times 10^{-3}$. Specifically, PANTHER was built with a **2-layer**, **1-head** configuration. All baseline models were configured identically to PANTHER.

These experiments followed the same training pipeline as WeChatPay, with hierarchical tokenization and discretization applied to transaction attributes. Given the smaller dataset sizes, training was completed within a **shorter time frame** while preserving model scalability.

## B.3 Tokenization of benchmark datasets

For the CCT dataset, user behavior tokens are constructed from payment amounts, payment methods, and merchant categories, resulting in a vocabulary of 16,847 tokens. User profiles include available card and card-holder information such as card brand, card type, and user age. For the MBD-mini dataset, user behavior tokens are derived from transaction attributes including amount, currency, event type, and the source and destination types, yielding a vocabulary of 40,791 distinct tokens.

In the Yelp dataset, user behavior tokens are formed by combining a business's city, category, star rating, and review count, where continuous features are bucketized. The original vocabulary of 40K tokens can be compressed to 17K tokens, covering 95% of all user-business interactions. User profiles consist of attributes such as the number of friends, number of reviews, and average star ratings. For the Movielens dataset, movie IDs are directly used as behavior tokens.

## B.4 Efficiency Evaluation

We compare GPU memory usage and inference time of the SPRM against the Transformer baseline across increasing sequence lengths, in Table 6. These experiments illustrate how SPRM scales more efficiently in both memory consumption and latency, particularly when handling longer sequences where the Transformer model fails due to memory overflow.

Table 6: Efficiency comparison between Transformer and SPRM across different sequence lengths

| Sequence Length | Transformer Memory(GB) | Transformer Inference Time(s) | SPRM Memory(GB) | SPRM Inference Time(s) |
|---|---|---|---|---|
| 1024 | 8.4 | 72.2 | 1.9 | 65.4 |
| 2048 | 29.7 | 113.9 | 3.2 | 70.3 |
| 4096 | OOM | - | 5.7 | 74.1 |
| 8192 | OOM | - | 10.8 | 93.8 |

## C    Transferability of PANTHER

We examine PANTHER 's capacity to transfer learned representations to new users, new datasets, and new domains. The experiments showcase the advantage of generative pretraining in label scarcity and cross-domain scenarios.

Table 7: Comparison of PANTHER with cold-start and user transfer settings on WeChatPay.

| | WeChatPay | | | |
|---|---|---|---|---|
| | HR@1 | HR@5 | HR@10 | HR@50 |
| Cold-Start | .0581 | .0834 | .0963 | .1308 |
| **User-Transfer** | **.2332(+301%)** | **.4494(+439%)** | **.5417(+463%)** | **.7280(+457%)** |

### C.1    User-Level Transferability

We pre-train PANTHER on one set of WeChatPay users and evaluate on another disjoint set for a cold-start recommendation setting. As shown in Table 7, the pre-trained PANTHER significantly outperforms the cold-start baseline, demonstrating its ability to preserve learned behavioral patterns and adapt to new users with minimal retraining. This finding is crucial for financial applications where new users frequently arrive and labeled data are sparse.

### C.2    Data-Level Transferability

Table 8: Experimental results of PANTHER on CCT and MBD-mini datasets after pretraining on WeChatPay dataset. The values in parentheses indicate the relative improvement compared to directly finetuning on these datasets

| Dataset | HR@1 | HR@5 | HR@10 | HR@50 |
|---|---|---|---|---|
| MBD-mini | .0539(+16.66%) | .1980(+10.43%) | .3143(+6.72%) | .7123(+3.82%) |
| CCT | .0248(+13.76%) | .0729(+10.79%) | .1050(+6.59%) | .2706(+4.08%) |

To assess cross-dataset adaptability, we pre-train PANTHER on WeChatPay and fine-tune it on MBD-mini and CCT. Table 8 shows an average HR@1 improvement of 16.66%, demonstrating effective transfer of our generative pretraining across diverse transaction contexts. The model retains valuable cross-data signals, such as user spending patterns, even when the merchant or product space changes.

## D    Ablation Experiments

To examine the sensitivity of PANTHER to the balance between $\mathcal{L}_{\text{gen}}$ and $\mathcal{L}_{\text{CL}}$, we vary the loss coefficient $\lambda$ and report the corresponding performance. The results are summarized in Table 9.

Table 9: Performance on WeChat Pay under different values of the loss coefficient $\lambda$.

| $\lambda$ | HR@1 | HR@5 | HR@10 |
|---|---|---|---|
| 0.1 | 0.2452 | 0.4837 | 0.5647 |
| 0.2 | 0.2441 | 0.4862 | 0.5644 |
| 0.4 | 0.2435 | 0.4869 | 0.5653 |
| 0.8 | 0.2430 | 0.4832 | 0.5629 |

