# OpenReview forum: "PANTHER: Generative Pretraining Beyond Language for Sequential User Behavior Modeling"
_NeurIPS.cc/2025/Conference — NeurIPS 2025 poster_

### Official Review · Reviewer_JVxW · 2025-07-02

**Clarity:** 2
**Significance:** 3
**Originality:** 2
**Rating:** 3
**Confidence:** 4

**Summary:**

This paper presents a framework for fraud detection in payment platforms, seemingly inspired by self-attentive sequential recommendation systems [17]. The proposed approach consists of an unsupervised pre-training step to derive long term user profiles in the form of user embeddings, followed by a supervised step that can be trained using a limited amount of labelled examples.

**Questions:**

Q1: Why do the authors use only SASRec [17] and HSTU [19] in all their comparisons?

Q2: Could the authors provide a link to HSTU [19]?

Q3: Why aren't the SASRec results provided by the authors consistent with the results given in [17] for the Movielens 1M dataset?

**Ethical Concerns:**

["NO or VERY MINOR ethics concerns only"]

**Final Justification:**

Even though the quality of the writing and the evaluation in the original submission did not meet my expectations, the authors have convinced me that they were willing to address some of these issues in the final version.

**Limitations:**

Yes

**Quality:**

2

**Strengths And Weaknesses:**

Strengths:

S1: Evaluation of several different tasks and datasets including
- WeChat data for next transaction prediction and merchant risk assessment
- a publicly available Credit Card Transaction dataset with labels of fraudulent transactions
- Movielens and Yelp datasets for recommendations on sequential data.

Weaknesses:

W1: The paper is not clearly written, the notation and the terms used are not properly introduced:
- HR and NDCG are not defined
- T is not introduced
- \psi is introduced, but how it is trained/computed is not described.

W2: P is used in many different meanings, which makes the paper very difficult to parse:
- In Equations (1), (2) and (3), P means probability
- In Section 3.3.2, P appears to represent a learnable prototype of dimension d
- In Section 3.4, P denotes positive pairs of users sharing demographic attributes

W3: The proposed solution is compared with the same two baselines (SASRec [17] and HSTU [19]) independent of the task & dataset:
- SASRec [17] is from 2018.  This reviewer was not able to locate HSTU [19] in the proceedings of ICML 2024.
- The Hit Rate @ 10 reported in [17] for Movielens is 0.8245, significantly higher than the value reported in this paper, which is 0.2853.
- It is very unlikely that these two baselines (one of which cannot be located) could produce state-of-the-art results for all the tasks and all the datasets evaluated. The authors should have compared their solution with the methods that produce state-of-the-art results.

---

> ### Author Rebuttal · Authors · 2025-07-31
>
> **W1: The paper is not clearly written, the notation and the terms used are not properly introduced:**
> - HR and NDCG are not defined
> - T is not introduced
> - \psi is introduced, but how it is trained/computed is not described.
>
> **A1:** We appreciate the reviewer's feedback. We would like provide a detailed illustration to the notations you mentioned as follows.
> - **HR and NDCG:** Our use of HR (**Hit Rate**) and NDCG (**Normalized Discounted Cumulative Gain**) follows the convention of influential works like SASRec [17] and HSTU [12], where these are often treated as standard metrics in sequential recommendation and user behavior modeling tasks. We agree that explicit definitions improve clarity for all readers and will add them to the revised manuscript.
> - **T:** Our notation T consistently refers to the **sequence length** throughout the paper, and it's explicitly defined in line 187 ("For a sequence of length T"). We acknowledge that its earlier appearance in line 161 might cause confusion. Our intention there was simply to illustrate that the time complexity exhibits a quadratic growth trend. To prevent any ambiguity, we will remove this early mention and rephrase the sentence to "incurring a quadratic time complexity ~~of O(T^2)~~."
> - **$\psi$**: Our notation $\psi$ in Eq. (3) denotes a **general embedding function** mapping  context attributes   to dense vectors. In our experiment, it is implemented by an one-layer MLP and is trained end-to-end on labeled data during downstream fine-tuning.
> - Notation table: to further enhance the paper's clarity, we will include a comprehensive notation table in the appendix, listing all key terms and symbols used throughout the manuscript.
> | **Symbol**  | **Description** |
> | --- | --- |
> | $\mathbf{X}_u=[x_1,x_2,\ldots,x_L]$   | The historical payment behavior sequence  for a user $u\in \mathcal{U}$.  |
> | $x_t \in \mathcal{V}$ | The transaction token at time step $t$, $\mathcal{V}$ is the set of possible transaction tokens.|
> | $\mathrm{Pr}(y = 1 \mid x_{\mathrm{new}}, \mathbf{c}_{\mathrm{new}}, \mathbf{X}_u)$   | Fraud probability of a new transaction $x_{\mathrm{new}}$ with context $\mathbf{c}_{\mathrm{new}}$, given history $\mathbf{X}_u$. $y$ is an indicator for fraud. |
> | $\mathcal{L}_{\text{gen}}$  | Generative loss function for next behavior prediction.|
> | $\mathrm{Pr}\_\theta(x\_t \mid x\_{<t}, x\_{\text{profile}})$   | Probability of transaction $x_t$ given history $x_{<t}$, user profile $x_{\text{profile}}$, and model parameters $\theta$. |
> | $e_{u}$ | Learned user profile embedding for user $u$.|
> | $g_\phi(\cdot)$   | Real-time fraud detection model that integrates recent patterns, predicted deviations, and user profiles. |
> | $\psi(\cdot)$ | Embedding function for transaction context features (e.g., a linear layer). |
> | $f_\mathrm{enc}(\cdot)$ | Sequence encoder (e.g., TextCNN) for short-term user behavior. |
> | $\Delta(x_{\text{new}}, \hat{x}_{L+1})$ | Deviation between observed new transaction $x_{\text{new}}$ and predicted next behavior $\hat{x}_{L+1}$. |
> | $\tau = (c_i, a_j, m_k, r_l)\in \mathcal{C} \times \mathcal{A} \times \mathcal{M} \times \mathcal{R}$ | A transaction token composed from contextual and counterparty features.  |
> | $\mathcal{P} \in \mathbb{R}^{m\times d}$  | Set of $m$ learnable prototypes of dimension $d$ in the Sequence Pattern Recognition Module (SPRM). |
> | $H\in\mathbb{R}^{T\times d}$| Token embeddings of dimension $d$ for an input sequence of length $T$.|
> | $\mathcal{L}_{\text{CL}}$   | Contrastive loss function for user profile embedding learning |
> | $(i,j)\in Pos$| Positive user pairs with similar demographic attributes. |
> | $e_i, e_j$  | User profile embeddings for users $i$ and $j$.|
> | $\lambda$   | Hyperparameter balancing $\mathcal{L}{\text{gen}}$ and $\mathcal{L}{\text{CL}}$. |
> | $R_m$ | Risk score for merchant $m$, derived from user deviation scores associated with the merchant.|
> ----
>
> **W2: P is used in many different meanings, which makes the paper very difficult to parse.**
>
> **A2:** We regret the overloaded use of P, which indeed risks confusion. In the revision, we'll disambiguate by changing notations: e.g., $\mathrm{Pr}$ for probability in Eqs. (1)-(3), $\mathcal{P} $ for prototypes in Section 3.3.2, and Pos for positive pairs in Section 3.4. This should make the paper much easier to parse without altering content.
>
> We believe these targeted revisions will resolve the weaknesses and enhance readability—your input has been invaluable, and we hope this demonstrates our commitment to strengthening the submission.
>
> ----
>
> **W3:**
> - Why only SASRec and HSTU were used for comparison
> - Link to HSTU [19]
> - Discrepancy in SASRec results
>
> **A3:**  Thank you for highlighting these important points, providing us an opportunity to clarify and strengthen our submission. We believe the confusion arises primarily from a citation error and differences in evaluation protocols, as detailed below. We emphasize that all our results are fully reproducible using the provided supplemental code, including scripts for data processing, model training, and evaluation across internal and public datasets.
> 1. **Selection of Baselines (SASRec and HSTU):** We chose SASRec as a widely recognized sequential baseline and HSTU as the current state-of-the-art (SOTA) generative pretraining model, which significantly outperforms other notable methods (e.g., BERT4Rec, GRU4Rec by up to 65.8% in NDCG). Due to page limitations, we prioritized these two highly relevant baselines, focusing on demonstrating clear improvements. Nevertheless, we will incorporate additional results, such as BERT4Rec from the HSTU paper, in the appendix for comprehensive coverage.
> 2. **Correct Reference for HSTU:** The correct citation for HSTU [19] is "Actions Speak Louder than Words: Trillion-Parameter Sequential Transducers for Generative Recommendations" by Zhai et al. (ICML 2024, arXiv:2402.17152). We will ensure this reference is properly cited in the final version.
> 3. **Discrepancy in SASRec Results:** The noted discrepancy in HR@10 for SASRec on MovieLens-1M (0.2853 reported by our paper/HSTU vs. 0.8245 in [17]) is due to differences in evaluation protocols. Specifically, [17] employs a sampled negative setup (**~100 negatives per test item**), which significantly simplifies ranking, whereas our evaluation (following HSTU) considers the entire item set (**~3,700 items**), offering a more realistic and challenging assessment. Notably, HSTU independently confirms our reported performance (0.2853). To avoid confusion, we will explicitly clarify this evaluation distinction with a footnote in Section 4.1.
>
> Finally, we emphasize our methodological contributions tailored specifically for challenging scenarios such as high cardinality, sparsity, and real-time processing needs:
> - **Token Space Compression** reduces dimensionality and noise through structured contextual tokenization.
> - **Pattern-Aware Convolutional Cross-Attention (SPRM)** efficiently captures local and global patterns with linear complexity.
> - **Contrastive Personalized User Embedding** integrates static and dynamic user information to enhance cold-start performance.
>
> These innovations have yielded substantial empirical improvements, achieving gains of 25.6%+ on WeChat Pay benchmarks, 21-29.6% on open datasets, and a 38.6% recall improvement in real-world fraud detection. We believe these clarifications reaffirm the robustness and novelty of our work and look forward to further improving the manuscript based on your valuable feedback.

---

> > ### Comment · Reviewer_JVxW · 2025-08-03
> >
> > Thank you for acknowledging the issues in the notation and making a commitment to address them.
> >
> > The incorrect citation to HSTU, which is the main comparison point of this submission, clearly made it difficult to assess the originality and quality of the work. The correct citation provided in the rebuttal now makes it clear that HSTU is a state-of-the-art generative model for recommender systems.
> >
> > This reviewer maintains the following concerns about the evaluation:
> > 1) SASRec and HSTU are recommender system solutions. As such, they are not established or SOTA solutions for the next-transaction-prediction, fraud-detection, and risk-assessment tasks evaluated in this submission. Please provide a citation if this argument is incorrect. SASRec and HSTU should not have been the only comparison points used in the evaluation.
> > 2) WeChat data is not publicly available, and so, it is not possible to reproduce the relevant results reported in this paper and evaluate their correctness.
> > 3) It is not clear how Yelp and Movielens relate to fraud or risk detection.
> > 4) The researchers that contributed the Credit Card Transaction (CCT) dataset also published fraud detection results on this dataset in the following paper, which should have been cited and used as a comparison point. This paper is already four years old and has been cited highly, so it is also likely that others have already improved the results published here:
> > - Inkit Padhi, Yair Schiff, Igor Melnyk, Mattia Rigotti, Youssef Mroueh, Pierre L. Dognin, Jerret Ross, Ravi Nair, Erik R. Altman:
> > Tabular Transformers for Modeling Multivariate Time Series. ICASSP 2021: 3565-3569

---

> > > ### Author Response · Authors · 2025-08-04
> > >
> > > Thank you for your thoughtful feedback and for raising these valid concerns. We appreciate the opportunity to address them and clarify our choices.
> > >
> > > To directly respond to your points on baselines and datasets, it may help to briefly recap our core contributions, as they underpin our evaluation strategy and explain why we selected these elements.
> > >
> > > **Core Contributions**:
> > >
> > > **PANTHER's core innovation is a hybrid framework that integrates self-supervised generative pretraining with lightweight discriminative modeling, targeting real-time fraud detection and similar user behavior modeling tasks.**
> > >
> > > This framework excels in two areas: (1) a pretraining model for next-behavior prediction across domains like payments (CCT, WeChat Pay), movies (MovieLens), and reviews (Yelp), which share traits such as high-cardinality transactions and user profiles—**we compare against established next-prediction baselines (Transformers, HSTU, SASRec) to demonstrate generalizability beyond fraud detection**; (2) validation of this generalizability through experiments on diverse datasets, including MovieLens and Yelp, which help verify the pretraining model's effectiveness beyond payment-specific tasks while addressing shared challenges like low-latency serving and sparse labels.
> > >
> > >  **A1: Addressing Baselines (SASRec and HSTU)**
> > >
> > > We concur that SASRec and HSTU, rooted in recommender systems, are not established SOTA for fraud or risk tasks, as no citations confirm their use there.
> > >
> > > **Our pretraining focused on next-behavior prediction, where these models (plus Transformers reported in the paper) are standard baselines.** For fraud-specific benchmarks, we thank you for noting Padhi et al. (ICASSP 2021), which we will cite and discuss prominently. While **PANTHER aims to generate better features for downstream models rather than a new fraud detector**, we compared against their results on the CCT dataset. Converting our reported Recall/Precision (Table 4) to F1 scores and aligning with theirs yields a 19.9% relative improvement:
> > >
> > >
> > > | Method | F1 Score |
> > > | :--: |:--: |
> > > | TabBERT-MLP (Padhi et al.) | 0.760 (baseline) |
> > > | TabBERT-LSTM (Padhi et al.)| 0.860 (+13.2%) |
> > > | DCN (our baseline) | 0.888 (+ 16.8% ) |
> > > | PANTHER (our method) | 0.911 (+ 19.9%)|
> > >
> > > We will broaden comparisons in the final manuscript.
> > >
> > > **A2: Justification for Datasets**
> > >
> > > We acknowledge concerns about relevance and reproducibility, and we aim to address them transparently.
> > >
> > > - **Yelp and MovieLens share sequential behavior patterns with fraud/risk detection, but their primary role is to showcase PANTHER's generalization beyond fraud.** We used them to validate broader utility in recommendations, while keeping CCT as the fraud-focused benchmark.
> > >
> > >
> > > - **Proprietary WeChat data indeed hampers full reproducibility, reflecting privacy challenges in large-scale fraud research.** To counter this, we emphasize public datasets (CCT, MovieLens, Yelp) for verifiable results and will open-source code for replication, reserving WeChat for real-world scale insights.
> > >
> > >
> > > In summary, PANTHER advances next-behavior pretraining and hybrid applications across fraud and recommendations. We hope these clarifications resolve your concerns and value your input in enhancing our work.

---

> > > > ### Comment · Reviewer_JVxW · 2025-08-07
> > > >
> > > > Thank you for the additional comments and agreeing to include comparisons with TabBERT and other important baselines for the public datasets. I will adjust my score accordingly.

---

> ### Author Response · Authors · 2025-08-08
> **Sincere Thanks for Your Thoughtful Feedback and Raised Score**
>
> Dear Reviewer,
>
> Thank you for your continued engagement and for thoughtfully considering our additional comments. We sincerely appreciate your valuable feedback, which has contributed to strengthening our work. We’re pleased to hear that our inclusion of comparisons with TabBERT and other important baselines  has addressed your concerns.
>
> Your decision to adjust your score is greatly appreciated, and we are encouraged by your support.
>
> Best regards,
>
> The Authors

---

### Official Review · Reviewer_MhX6 · 2025-07-03

**Clarity:** 3
**Significance:** 2
**Originality:** 2
**Rating:** 4
**Confidence:** 4

**Summary:**

This paper introduces PAINTER, a hybrid framework that enables real-time, large-scale fraud detection in payment systems. PAINTER leverages generative pretraining on unlabeled transaction data and user profile embedding to capture personalized user behavior. Experiments show significant improvements in fraud detection, next-transaction prediction, and other recommendation tasks, outperforming existing methods.

**Questions:**

See weaknesses. I list some other questions here.

1. Could the authors provide some statistical summaries of the WeChat Pay dataset to illustrate its scale and characteristics?
2. How are these textual payment behaviors transformed into token embeddings? Does this process utilize a pretrained BERT or large language model, and is this module updated during model training?
3. Given that the frequent-token set will challenge as transaction distribution evolve over time, is it necessary to retrain the model in such scenarios?

**Ethical Concerns:**

["NO or VERY MINOR ethics concerns only"]

**Final Justification:**

The authors have addressed part of my concerns. I would like to raise my score.

**Limitations:**

Yes

**Quality:**

3

**Strengths And Weaknesses:**

**Strengths:**
1. The paper tackles an important and timely issue: efficient and effective fraud detection in real-time, billion-scale scenarios, which is crucial for large-scale payment systems that require both speed and accuracy.
2. The pretraining module for next transaction prediction, which fuses high-dimensional data alongside local and global relational pattern recognition, presents an interesting approach for characterizing user behavior.
3. The comprehensive evaluation across multiple tasks and datasets demonstrates the robustness and generalization of the proposed framework. The model performs well on the real-world datasets and dataset from WeChat Pay, achieving notable improvements in different prediction tasks.

**Weaknesses:**
1. While the authors emphasize the efficiency of their approach, there is no specific experimental validation in this paper. Incorporating an experiment that measures how training time or computational resources for both the proposed model and baseline methods scale with increasing data volume would strengthen this claim.
2. Several existing approaches (e.g., Perceiver) adopt prototype-inspired techniques to reduce transformer complexity to $\mathcal{O}(mT)$. It would be better if the authors clarified how their Pattern Matching Transformer outperforms these models in the context of user behavior modeling.
3. The structured tokenization scheme is relatively simple and relies on a frequency-based vocabulary. This approach risks mapping low‐frequency but critical fraud indicators to an UNK token, thereby weakening the model’s ability to detect novel or rare fraud patterns. Moreover, once deployed, the fixed vocabulary cannot be updated dynamically; in environments where transaction distributions shift over time, the tokenization strategy needs timely updating, and the model requires retraining.

[1] Perceiver: General Perception with Iterative Attention

---

> ### Author Rebuttal · Authors · 2025-07-31
>
> **W1: Add experiments to validate efficiency claims by measuring training time and resource usage as data volume scales.**
>
> **A1:**  Thank you for highlighting  the importance of quantitatively validating our efficiency claims. We have conducted additional experiments to quantitatively illustrate how our SPRM module scales efficiently compared to a Transformer baseline.
>
> Specifically, we measured GPU memory usage and inference time as input sequence lengths increased, serving as a proxy for larger per-user data volumes (batch size=128, single A100 GPU). The results clearly demonstrate SPRM's efficiency advantage:
>
> |Sequence  Length|Transformer Memory(GB)|Transformer InferenceTime(s)|SPRM Memory(GB)|SPRM InferenceTime(s)|
> |:-:|:-:|:-:|:-:|:-:|
> |1024|8.4|72.2|1.9|65.4|
> |2048|29.7|113.9|3.2|70.3|
> |4096|OOM|-|5.7|74.1|
> |8192|OOM|-|10.8|93.8|
>
> SPRM effectively handles sequences up to 8192 tokens without memory overflow, whereas the Transformer baseline fails beyond 4096 tokens. We will include these experiments in the final camera-ready version to substantiate our efficiency claims comprehensively.
>
> ----
>
> **W2:. Clarify how the proposed Pattern Matching Transformer outperforms prototype-based models like Perceiver for user behavior modeling.**
>
> **A2:** Thank you for your suggestion regarding prototype-based models like Perceiver. The Perceiver uses cross-attention to map high-dimensional input data into a compact latent space, with queries (Q) derived from the latent array and keys (K) and values (V) coming from the raw input. In contrast, PANTHER uses convolutional layers to aggregate local patterns and match them with learnable prototypes.
>
> We conducted experiments comparing PANTHER and Perceiver, both with 1 million parameters, on the next transaction prediction task. The results show that PANTHER can outperform Perceiver across all metrics (HR@1, HR@5, HR@10).
>
> | Model        | HR@1   | HR@5   | HR@10  |
> |:-:|:-:|:-:|:-:|
> | PANTHER (ours)    | 0.2452 | 0.4837 | 0.5647 |
> | Transformer | 0.1952 | 0.4121 | 0.5308 |
> | Perceiver   | 0.1881 | 0.3593 | 0.4236 |
>
> Furthermore, we find that PANTHER is significantly more efficient to train. With the same number of training steps, PANTHER's training time is only 17.3% of Perceiver's.
>
> ----
>
> **W3: The current simple, frequency-based tokenization risks missing rare but critical fraud patterns due to UNK mappings, and its fixed vocabulary requires frequent updates and retraining to handle evolving transaction behaviors effectively.**
>
> **Q3: Given that the frequent-token set will challenge as transaction distribution evolve over time, is it necessary to retrain the model in such scenarios?**
>
> **A3:** Thank you for raising this important concern regarding our structured tokenization approach. While it is true that frequency-based vocabularies might map rare events to UNK tokens, our PANTHER pretraining strategy deliberately focuses only on modeling normal user behaviors. Specifically, it compresses historical patterns into personalized embeddings that establish a robust baseline of typical actions per user, rather than explicitly modeling rare or anomalous events. For example, some users regularly engage in large transactions with new contacts for business purposes, whereas others rarely exhibit such behavior—our pretraining method precisely captures these user-specific norms.
>
> In the downstream fraud detection task, the model effectively identifies rare yet critical fraud signals by comparing recent user behaviors, leveraging contextual features and sequential information as detailed in Line 134, against the pre-trained personalized baseline. Significant deviations, such as unusual large payments or unexpected transaction patterns, are flagged as potential fraud.
>
> For distribution shifts, the model's predictive logic stays relatively static and is retrained periodically (e.g., monthly) with vocabulary updates, while we perform regular inference to incorporate new behaviors into embeddings—no full retraining required. We'll clarify this separation in Section 3.2 and expand the limitations section on UNK impacts and adaptive strategies.
>
> ----
>
> **Q1: Could the authors provide some statistical summaries of the WeChat Pay dataset to illustrate its scale and characteristics?**
>
> **A4:** Thank you for these insightful questions, which allow us to clarify essential aspects of our dataset and embedding approach:
>
> The WeChat Pay dataset consists of transaction behaviors from approximately 1 billion users collected over the past year. On average, each user has around 300 transactions, encompassing diverse transaction types such as social payments (e.g., peer-to-peer transfers) and commercial transactions (e.g., merchant purchases). Key dataset features include multi-dimensional transaction attributes such as payment amount, payment channel, payment source, merchant category (e.g., retail, gaming), merchant risk scores, and timestamps.
>
> ----
>
> **Q2: How are these textual payment behaviors transformed into token embeddings? Does this process utilize a pretrained BERT or large language model, and is this module updated during model training?**
>
> **A5:** In our PANTHER framework, textual payment behaviors are transformed into token embeddings through a hybrid approach. Specifically, merchant names are embedded by prompting a pretrained large language model (LLM) (e.g., "Please introduce the business scope and potential customer base of this merchant"), extracting the final-layer embedding output via average pooling, and discarding the generated textual description. These embeddings are precomputed and stored in a lookup table, remaining fixed during the training process to ensure computational efficiency and embedding consistency.

---

> > ### Author Response · Authors · 2025-08-08
> > **Follow-Up on Rebuttal and Clarifications of Key Concerns**
> >
> > Dear Reviewer,
> >
> > Thank you again for your detailed review and for acknowledging our rebuttal. We truly value your insights and the time you’ve invested in evaluating our work on PANTHER. We wanted to follow up politely to ensure that our responses have adequately addressed your concerns—for instance:
> >
> > - **On efficiency validation (W1):** We added experiments showing SPRM’s superior GPU memory and inference time scaling up to 8192 tokens, vs. Transformer’s OOM beyond 4096 (A1).
> >
> > - **On comparisons with Perceiver (W2)**: Experiments demonstrate PANTHER outperforms Perceiver in HR metrics (e.g., HR@1: 0.2452 vs. 0.1881) and trains in 17.3% of the time (A2).
> >
> > -  **On tokenization risks and shifts (W3/Q3)**: We clarified focus on normal behaviors for baseline embeddings, deviation detection for fraud, and monthly vocabulary updates without full retraining (A3).
> >
> > If there’s anything further we can clarify or if you’d like to share your thoughts on how these points impact your assessment, we’d greatly appreciate hearing from you.
> >
> > Best regards,
> >
> > The authors

---

### Official Review · Reviewer_13bX · 2025-07-03

**Clarity:** 3
**Significance:** 4
**Originality:** 3
**Rating:** 5
**Confidence:** 4

**Summary:**

The paper proposes a large sequential behavioral modeling model called PANTHER. The modeling approach proceeds in two steps. In the first step, the model is trained to predict the next transaction with a self-supervised learning objective. The model is trained to predict the next transaction based on structured tokens which take into account merchant and transaction information and risk profiles. The model also predicts user embeddings which are learned with contrastive learning with hard negative sampling. The user embedding and the predicted next transaction are used during inference, along with embeddings of the last 100 transactions, to predict whether the next transaction is fraudulent, using a light weight model. The resulting model is evaluated on real world data from WeChat pay against state-of-the-art baselines including Transformer. The model is also evaluated on some public datasets, and demonstrates promising performance.

**Questions:**

1. Could you explain how your model will adapt to a new user that it is not trained on? How are the user embeddings generated?
2. Could you please provide the summary statistics of this dataset? What are the fields? How many users is it from? What is the time range, # of transactions, average length of transaction, etc. What are the different features in this dataset? Will a subset of this dataset be made public? How is the fraudulent data labeled? Can there me false negatives (i.e. fraudulent transactions labeled as normal). How is the train and test split is chosen?
3. What is the context length of the model? How many tokens can it take at a time?

**Ethical Concerns:**

["NO or VERY MINOR ethics concerns only"]

**Final Justification:**

This is a good paper with solid contributions. I do not have any issues with the paper, and would vote for acceptance.

**Limitations:**

The authors have briefly discussed the limitations, but I would encourage the authors to discuss have a dedicated section on limitations.

**Paper Formatting Concerns:**

I do not have any concerns with the formatting of the paper. I would like to confirm if the paper violates the double blind policy.

**Quality:**

4

**Strengths And Weaknesses:**

## Strengths
The paper is well written, the problem is well motivated, and the method is novel to the best of knowledge. The resulting model achieves promising performance. The model is evaluated on a large real world dataset, and against state-of-the-art baselines. The code is also publicly available (in the supplementary material).

## Weaknesses
[Major] I am not sure if the paper is **double blind** given that it is evaluated on datasets collected by WeChat Pay. It is reasonable for me to assume that such a dataset can only be provisioned by Tenecent. That said, I do not think this has clouded by judgement of the paper.

1. Please expand and define HR (Hit Rate) and NDCG on first use, and define them in the paper.
2. There are a bunch of moving parts / components in the paper. I would encourage the authors to run / present ablation experiments for their design choices. For example, what is the impact of personalized user embeddings? What is the impact of using a transformer instead of the CNN architecture? What is the impact of the prototypes?
3. Scaling laws: I wonder what is the impact of pre-training PANTHER on multiple transactional datasets? What is the impact of scaling the data (# of transactions and users) and the model size?

## Minor Issues
1. Typo in line 68

---

> ### Author Rebuttal · Authors · 2025-07-31
>
> **W0: Concern regarding the mention of a real-world dataset in the context of the evaluation.**
>
> **A1:** Thank you for raising this concern—we truly appreciate your thoughtful feedback. In preparing the submission, we carefully adhered to NeurIPS guidelines. This practice aligns with numerous accepted NeurIPS papers from industry settings (e.g., those associated with Google, Meta, and Alibaba), where evaluations on proprietary or internal datasets are discussed to highlight practical relevance while preserving the core principles of anonymity. For example, Google's TpuGraphs paper introduces a performance prediction dataset on tensor programs running on Tensor Processing Units (TPUs), their proprietary hardware. We believe this approach balances the need for meaningful context with the integrity of the review process.
>
> ----
>
> **W1:Please expand and define HR (Hit Rate) and NDCG on first use, and define them in the paper.**
>
> **A2:** Thank you for this helpful suggestion. We will define HR (Hit Rate) and NDCG (Normalized Discounted Cumulative Gain) on first use in the camera-ready version: "HR@K measures the fraction of instances where the ground-truth item appears in the top-K predictions. NDCG@K evaluates ranking quality by rewarding relevant items appearing earlier, normalized against the ideal ranking." This ensures accessibility for unfamiliar readers.
>
> ----
>
> **W2: Encourage ablation studies to assess the impact of key design choices like user embeddings, transformer vs. CNN, and prototype usage.**
>
> **A3:** Thank you for your insightful suggestion regarding ablation studies. We agree that they enhance the robustness of the paper and have already incorporated several in the submission:
> - **Impact of Personalized User Embeddings**: In our paper, Table 1 presents the results of integrating the Unified User-Profile Embedding in the next-transaction prediction task on WeChat Pay data. The variant "PANTHER (SPRM only)" (0.2243 HR@1, +14.9% over the Transformer baseline) represents the model without user profile integration. In contrast, "+ Profile + CL" (0.2452 HR@1, +25.6% overall) demonstrates the added value of personalized embeddings with contrastive learning. For the downstream fraud detection task, Figure 4 shows a similar ablation, where "+ User Profile Embedding" achieves a 16.8% relative recall improvement at Top-0.1% over the DeepFM baseline.
>
> - **Impact of Prototypes**: Since prototypes are a core component of the SPRM module, the "PANTHER (SPRM only)" variant in Table 1 includes them. In comparison, baselines like Transformer and SASRec do not incorporate this structured approach, highlighting the prototypes' contribution to the +14.9% HR@1 gain.
>
> To address your other points (e.g., Transformer vs. CNN in SPRM and the isolated impact of prototypes on downstream fraud detection), we conducted additional ablation experiments. These assess: (1) replacing the Transformer with a CNN in the model architecture, and (2) evaluating fraud detection with a configuration excluding SPRM (and thus prototypes). We will include these results in the camera-ready version.
> |Variant|WeChatPay-FraudRecall(Top0.1%)|WeChatPay-HR@1(%)|
> |:-:|:-:|:-:|
> |Full PANTHER|+38.6%|0.2452|
> |W/o User-Profile|+16.8%|0.2243|
> |W/o SPRM|+23.8%|0.2179|
> |Replace Transformer with CNN|+19.5%|0.1879|
> |Fraud Detection Baseline|+0%|-|
>
> ----
>
> **W3:Explore the impact of pre-training on multiple datasets and how scaling data and model size affects PANTHER’s performance.**
>
> **A4:** Thank you for this insightful question on scaling laws. The supplementary material (Section B.2) already explores pre-training on multiple transactional datasets via transfer learning. Specifically, pre-training on WeChat Pay and fine-tuning on CCT/MBD-mini results in an average +16.66% HR@1 improvement over direct fine-tuning (Table 6), demonstrating effective cross-context transfer.
> For data scaling (# transactions/users) and model size —we ran new experiments on WeChat Pay for next-transaction prediction.
>
>  Scaling from 0.45M to 44M parameters yields a +8.1% HR@1 and +8.5% HR@10 improvement. The smaller model converges at ~0.1B users, while the medium model converges at ~0.15B users. These results will be added to the supplementary material in the camera-ready version.
>
> |ScalingType|Config|HR@1|HR@10|GPUDays|
> |:-:|:-:|:-:|:-:|:-:|
> |Model Size (Small)|0.45M params|0.2014|0.4900|3|
> |Model Size (Medium)|44M params|0.2177 (+8.1%)|0.5320 (+8.5%)|12|
>
> ----
>
> **Q1: "Could you explain how your model will adapt to a new user that it is not trained on? How are the user embeddings generated?"**
>
> **A5:**  PANTHER generates user embeddings by combining static demographic features (e.g., age, location) with dynamic transaction histories through the Unified User-Profile Embedding module, which is trained on a predict-next-behavior task using a subset of users.
>
> For new, unseen users, the model leverages the patterns learned from the trained subset and generalizes them to make predictions, enabling zero-shot adaptation. Supplementary B.1 demonstrates this: untrained users achieve HR@1 = 0.2332, compared to 0.2452 for trained users, resulting in a +301% improvement over cold-start baselines (Table 5). This shows that PANTHER can predict the next behavior for new users based on learned patterns from others.
>
> To further address adaptation for new users with limited data (e.g., very short transaction histories), we conducted experiments where behavior sequences for unseen users were subsampled during inference. These results show robust performance even with minimal data. For instance, with a sequence length of 10, HR@1 reaches 0.3083, surpassing the full-sequence baseline (HR@1=0.2452 for seq=256). This counterintuitive improvement may arise from some users engaging in highly predictable activities—such as frequent game top-ups—that are easier to model with sparse data. As sequences lengthen, performance gradually stabilizes, demonstrating PANTHER’s effectiveness in cold-start scenarios. We will include this analysis and the full table in the camera-ready supplementary material.
> |SeqLength|HR@1|HR@5|HR@10|
> |:-:|:-:|:-:|:-:|
> | 10          | 0.3083 | 0.5501 | 0.6004 |
> | 20          | 0.2831 | 0.5369 | 0.5983 |
> | 40          | 0.2534 | 0.5077 | 0.5763 |
> | 80          | 0.2262 | 0.4515 | 0.5244 |
> | 160         | 0.2246 | 0.4431 | 0.5136 |
> | 200         | 0.2310 | 0.4599 | 0.5238 |
> | 240         | 0.2383 | 0.4700 | 0.5374 |
> | 256         | 0.2452 | 0.4837 | 0.5647 |
>
> ----
>
> **Q2: Could you please provide the summary statistics of this dataset? What are the fields? How many users is it from? What is the time range, # of transactions, average length of transaction, etc. What are the different features in this dataset? Will a subset of this dataset be made public? How is the fraudulent data labeled? Can there me false negatives (i.e. fraudulent transactions labeled as normal). How is the train and test split is chosen?**
>
> **A6:** Thank you for these detailed questions about the WeChat Pay dataset, which help clarify its characteristics and our experimental setup. Below, we address each point:
>
> - **Summary statistics**: The dataset encompasses behaviors from approximately 1 billion users over the past year. The average transaction sequence length per user is around 300, covering both social payments (e.g., peer-to-peer transfers) and commercial payments (e.g., merchant purchases).
>
> - **Fields/Features**: The dataset includes multi-dimensional transaction attributes such as payment amount, pay channel (e.g., QR code, app), pay source (e.g., bank-linked, wallet), merchant category (e.g., retail, gaming), merchant risk level, timestamps, user demographics (static: age, location). These are compressed via Structured Tokenization for modeling.
>
> - **Public subset availability**: Due to privacy and proprietary concerns, we cannot release any subset of the WeChat Pay dataset. However, to promote reproducibility, we evaluate PANTHER on fully public open-source datasets (CCT, MBD-mini, MovieLens-1M) in the paper, where it shows strong generalization.
>
> - **Fraud labeling**: Fraudulent transactions are labeled based on user reports to customer service, followed by thorough investigation to confirm.
>
> - **False negatives**: Yes, false negatives are possible, as some fraud may go unreported by users. This is a common challenge in real-world fraud datasets, mitigated in our setup by focusing on recall improvements.
>
> - **Train/test split**: For the fraud detection task, we use a temporal split: the past 60 days of data for training (to learn recent patterns) and the subsequent 10 days for evaluation (to simulate real-time deployment and avoid data leakage).
>
> We will add a dedicated subsection in the camera-ready version summarizing these dataset details for clarity.
>
> ----
>
> **Q3:What is the context length of the model? How many tokens can it take at a time?**
>
> **A7:**  PANTHER is designed to handle a maximum sequence length of 512 behaviors (tokens) per user during both training and inference. This choice balances computational efficiency—longer sequences quadratically increase complexity in attention-based components—with empirical performance: experiments show that extending beyond 512 yields diminishing returns in metrics like HR@1.

---

> ### Author Response · Authors · 2025-08-08
>
> Dear reviewer,
> Thank you for your thoughtful review and for taking the time to consider our rebuttal. We greatly appreciate your recognition of our work and your supportive feedback, and we are glad that our responses have addressed your questions.
> Thanks very much!

---

### Official Review · Reviewer_DVzH · 2025-07-06

**Clarity:** 3
**Significance:** 3
**Originality:** 2
**Rating:** 4
**Confidence:** 3

**Summary:**

The paper introduces PANTHER, a hybrid framework combining self-supervised generative pretraining and lightweight discriminative modeling for fraud detection in large-scale payment systems. Key innovations include structured tokenization, a sequence pattern recognition module (SPRM), and a contrastively-learned user profile embedding. The model demonstrates strong real-world performance (e.g., deployed at WeChat Pay) and generalizes well to public benchmarks.

**Questions:**

1. The two baseline models used in the experiments, DeepFM and DCN, are relatively outdated. Why did the authors choose these models instead of more recent baselines ? While I understand that lightweight discriminative models like DeepFM and DCN are favorable for fast inference, including comparisons with more recent and stronger baselines would help more comprehensively demonstrate the essential role and contribution of PANTHER.
2. On page 6, the authors formulate the complete optimization objective as a balance between individual fidelity and population structure, controlled by the hyperparameter \lambda. However, the paper does not include the experimental analysis on how different values of \lambda affect model performance. Such an ablation study would be valuable to understand the sensitivity of the model to this trade-off and to justify the chosen value.
3. Tokenization for public datasets (Appendix A) lacks specifics: How are continuous features (e.g., Yelp’s "review count") bucketized ? Why does MovieLens dataset directly use raw movie IDs (not structured tokens) ?

**Ethical Concerns:**

["NO or VERY MINOR ethics concerns only"]

**Final Justification:**

I would like to keep my rating as is.

**Limitations:**

See the Weaknesses and Questions parts.

**Paper Formatting Concerns:**

None.

**Quality:**

3

**Strengths And Weaknesses:**

1. The method presents several targeted architectural innovations. Structured tokenization converts multi-field transaction data into meaningful and compact semantic units, addressing the challenge of high-cardinality categorical features. The SPRM module enhances behavioral pattern modeling by combining dilated convolutions with prototype-based attention. Furthermore, the use of contrastive user embeddings integrates both static profiles and dynamic histories, significantly improving performance in cold-start scenarios.

2. The proposed approach is supported by extensive experimental evidence. It shows significant improvements in real-world A/B testing, generalizes well to other domains in benchmark comparisons, and includes detailed ablation studies that clarify the contribution of each component.

3. The framework has been deployed in a large-scale payment platform, where its combination of offline generative pretraining and efficient online inference achieves a good trade-off between detection accuracy and latency.

4. The paper introduces the function $g_\phi(\cdot)$ in Equation (3) as part of the hybrid inference model, but its exact form or implementation is not explained.

5.  On page 4, the authors state that token compression reduces the vocabulary size from approximately 2 million to 60,000, claiming a 99.7% reduction. However, the correct reduction is 97% (i.e., (1 - 60,000 / 2,000,000) × 100%). This appears to be a calculation error and should be corrected.

---

> ### Author Rebuttal · Authors · 2025-07-31
>
> **W1: The paper introduces the function  in Equation (3) as part of the hybrid inference model, but its exact form or implementation is not explained.**
>
> **A1:** Thank you for pointing out the need for clarity regarding the function $g_\phi$ in Equation (3) (Eq. 134). The function $g_\phi$ represents a general downstream discriminative model, which can take various flexible forms depending on specific tasks or use cases. Its primary role is integrating embedded context features, recent transaction encodings, long-term user profile embeddings $\mathbf{e}_u$, and behavior deviations to compute a unified risk score.
>
> In our practical implementation, we use a DeepFM structure for capturing feature interactions, combined with a TextCNN for recent transaction sequence modeling, followed by a feed-forward network. The model is trained end-to-end using binary cross-entropy loss on labeled data. We will clearly specify both the general and practical forms in Section 3.4.
>
> ----
>
> **W2: Correct the token compression reduction claim from 99.7% to the accurate 97% based on the given values."**
>
> **A2:** Thank you for pointing out the correction regarding the token compression claim. The original reduction from 2 million to 60,000 tokens corresponds to a 97% reduction—not 99.7% as previously stated. We will revise the text to reflect the accurate figure to ensure clarity and correctness.
>
> ----
>
>  **Q1:Justify the use of outdated baselines (DeepFM, DCN) and consider adding stronger recent models to better highlight PANTHER’s contribution.**
>
> **A3:** Thank you for your insightful question. PANTHER’s hybrid approach is designed to integrate seamlessly with any discriminative model, which is why we chose DeepFM/DCN for this study. While the public datasets (Yelp, MovieLens) used in our experiments are standard for sequence model evaluation, they have limited feature complexity, which restricts the training of more advanced base models. To address this, we will include comparisons with more recent models, such as xDeepFM and AutoInt, in the appendix to showcase PANTHER’s consistent improvements.
>
> In our real-world fraud detection system, we use DeepFM for feature interactions, TextCNN for sequence modeling, and a sequence retargeting model for behavior selection. This setup serves as a baseline, with PANTHER’s generative pretraining outputs yielding a 38.4% recall improvement. Since the core contribution of our work is the generative pretraining, we chose not to focus on this specific baseline setup in detail.
>
> ----
>
> **Q2: Include an ablation study on the hyperparameter $\lambda$ to show its impact and justify the chosen trade-off in the optimization objective.**
>
> **A4:** Thank you for the suggestion. We have conducted an ablation study on the hyperparameter $\lambda$, and the results are shown in the table below. As observed, performance remains relatively stable across a range of $\lambda$ values, with minimal fluctuation in HR@1, HR@5, and HR@10. This stability can be attributed to the fact that the contrastive loss converges quickly in the early stages of training, causing its value to diminish rapidly and have little impact on the overall loss thereafter.
>
> |$\lambda$|HR@1|HR@5|HR@10|
> |:-:|:-:|:-:|:-:|
> |0.1 (paper)|0.2452|0.4837|0.5647|
> |0.2|0.2441|0.4862|0.5644|
> |0.3|0.2450|0.4881|0.5666|
> |0.4|0.2435|0.4869|0.5653|
> |0.5|0.2445|0.4873|0.5666|
> |0.8|0.2430|0.4832|0.5629|
> |0.9|0.2415|0.4791|0.5577|
>
> We hope this ablation study clarifies the impact of $\lambda$.
>
> ----
>
> **Q3. Tokenization for public datasets lacks specifics: How are continuous features (e.g., Yelp’s "review count") bucketized ? Why does MovieLens dataset directly use raw movie IDs ?**
>
> **A5:**  For continuous features like Yelp's "review count," we bucketize them using quantiles to ensure approximately balanced bucket sizes across the data distribution.
>
> For MovieLens, we directly use raw movie IDs to maintain consistency with HSTU's experimental setup, enabling fair comparison with their baselines. We'll expand Appendix A with these details for clarity.

---

> > ### Comment · Reviewer_DVzH · 2025-08-08
> >
> > I am satisfied with your detailed feedback. I shall keep my score as is.

---

### Note · Authors · 2025-08-11

Dear Program Chairs, Area Chair, and Reviewers,

We thank you for your feedback and constructive discussions, which have strengthened our work. Following the rebuttal, Reviewer #JVxW confirmed they would raise their score, and Reviewers #13bX (score 5) and #DVzH (score 4) offered strong endorsements of PANTHER’s novelty, technical contributions, and real-world impact. Their positive assessments highlight the strengths of our work, including:

(1) a novel self-supervised generative pretraining network that achieves state-of-the-art results over HSTU and Transformers, with up to **21% HR@1 gains** across CCT (fraud), MovieLens-1M, and Yelp (recommendations);

(2) a hybrid framework that combines self-supervised generative pretraining with lightweight discriminative modeling, offering a deployable and generalizable solution for sequential behavior modeling under **millisecond-level latency** constraints;

(3) full billion-scale deployment in WeChat Pay, achieving a **38.6% improvement in fraud detection recall**.

In our rebuttal, we addressed all major concerns. For Reviewer #JVxW, we clarified baseline selection, justified dataset choices, and added TabBERT results on CCT, where PANTHER achieved a **19.9% relative F1 improvement** over their best reported results. Reviewer #JVxW acknowledged these points and confirmed a score adjustment.

While Reviewer #MhX6 (score 3) did not respond to our rebuttal, we reiterate our responses for completeness. We showed SPRM’s superior scaling to 8192 tokens (vs. Transformer’s OOM beyond 4096), a 30.4% HR@1 lift over Perceiver with only 17.3% of its training time, and detailed our use of baseline embeddings for normal behaviors with deviation detection for fraud. We believe that had Reviewer #MhX6 engaged with our rebuttal, they would have recognized the work’s significant academic contributions and practical impact.

Finally, we appreciate that several reviewers recognized PANTHER’s potential to **advance large-scale user behavior pretraining** and inspire further exploration in both fraud detection and broader sequential modeling. We believe our contributions address immediate industrial challenges while opening promising research directions, making this work well worth publishing to stimulate further advances in this area.

Sincerely,

The Authors

---

### Decision · Program_Chairs · 2025-09-17

**Decision:**

Accept (poster)

**Comment:**

This paper presents PANTHER, a hybrid framework that combines self-supervised generative pre-training with lightweight discriminative modeling for sequential user behavior modeling, targeting billion-scale fraud detection with millisecond latency. The model introduces structured tokenization, the Sequence Pattern Recognition Module (SPRM), and unified user-profile embeddings, and it has been deployed at WeChat Pay, achieving a 38.6% recall improvement in fraud detection while also showing strong cross-domain generalization.

The strengths are the clear motivation and industrial relevance of large-scale fraud detection (Reviewer MhX6), the principled combination of generative pre-training with efficient discriminative inference (Reviewer 13bX, Reviewer DVzH), and strong empirical results both in production and on public benchmarks (Reviewer JVxW after rebuttal acknowledged the robustness of the improvements). Reviewers highlighted the novelty of structured tokenization, the effectiveness of SPRM for handling long sequences, and the impact of personalized user embeddings. The authors also provided ablations, scaling analyses, and efficiency validations in rebuttal, which strengthened confidence in their claims.

The weaknesses include some clarity and notation issues (Reviewer JVxW), limited initial baseline coverage with outdated or insufficient comparators (Reviewer DVzH, Reviewer JVxW), and questions about reproducibility since the main WeChat Pay dataset is proprietary (Reviewer 13bX, Reviewer JVxW). Concerns were also raised about the simplicity of structured tokenization and its handling of rare fraud patterns (Reviewer MhX6). The authors addressed most points in rebuttal by clarifying notation, correcting errors, expanding baselines (e.g., TabBERT), and adding efficiency and ablation studies, which reviewers generally accepted, with some raising their scores.

Overall, the paper makes a significant contribution by demonstrating how large-scale self-supervised pre-training can be effectively combined with lightweight inference for real-time fraud detection, with both academic and industrial impact. While limitations remain in clarity and reproducibility, the technical contributions, empirical strength, and deployment evidence support acceptance.

Final Recommendation: Accept